# Proton Therapy in the Management of Pancreatic Cancer

**DOI:** 10.3390/cancers14112789

**Published:** 2022-06-04

**Authors:** Jana M. Kobeissi, Charles B. Simone, Haibo Lin, Lara Hilal, Carla Hajj

**Affiliations:** 1Department of Radiation Oncology, School of Medicine, American University of Beirut Medical Center, Beirut 1107, Lebanon; jmk24@mail.aub.edu (J.M.K.); lh54@aub.edu.lb (L.H.); 2Department of Radiation Oncology, New York Proton Center, New York, NY 10035, USA; csimone@nyproton.com (C.B.S.II); hlin@nyproton.com (H.L.); 3Department of Radiation Oncology, Memorial Sloan Kettering Cancer Center, New York, NY 10027, USA

**Keywords:** pancreatic cancer, proton radiation therapy, toxicity, pencil beam scanning, double scattering proton therapy, intensity modulated proton therapy

## Abstract

**Simple Summary:**

Radiation treatment plays a pivotal a role in the management of pancreatic cancer, mainly in the neoadjuvant setting for borderline resectable tumors and in the definitive setting for unresectable localized disease. Most of the studies on pancreatic cancer use photon-based radiation therapy modalities. However, there is a growing interest in the application of protons therapy for gastrointestinal cancers. This review summarizes the literature on the use of proton therapy in the management of pancreatic cancer.

**Abstract:**

Radiation therapy plays a central role in the treatment of pancreatic cancer. While generally shown to be feasible, proton irradiation, particularly when an ablative dose is planned, remains a challenge, especially due to tumor motion and the proximity to organs at risk, like the stomach, duodenum, and bowel. Clinically, standard doses of proton radiation treatment have not been shown to be statistically different from photon radiation treatment in terms of oncologic outcomes and toxicity rates as per non-randomized comparative studies. Fractionation schedules and concurrent chemotherapy combinations are yet to be optimized for proton therapy and are the subject of ongoing trials.

## 1. Introduction

Pancreatic cancer accounts for around 3.2% of all new cancer cases per year in the United States [1]. Even with improvements in treatment modalities, long-term survival is limited, and the 5-year relative survival rate is estimated at just 11.5%. Localized disease has a better prognosis, but pancreatic cancer is difficult to detect at an early stage, with more than half of the patients having metastatic disease at diagnosis [1].

Radiation therapy is one of the locoregional treatment modalities used to treat cancer. The most commonly used radiation modality is X-ray, also called photon therapy. While the energy deposited by X-rays peaks at a certain depth, it extends beyond that peak to potentially affect normal tissues [2]. Other forms of radiation use particles, such as protons. Given the properties inherent to such heavy particles, energy is deposited at a particular depth with minimal scattered dose delivered to nearby organs [3]. Different ways are used to shape proton beams, with scanning methods being more conformal than passively scattered ones. In fact, scanning methods have allowed for the administration of intensity-modulated proton therapy (IMPT) [3]. 

For pancreatic cancer, radiation is often given concurrently with chemotherapy, which acts as a radiosensitizer. While the exact mechanism is unclear, sensitization may be due to decreasing the proportion of cells in the S phase of the cell cycle, during which cells are more likely to resist DNA damage [4]. The radiosensitivity of pancreatic tumor cells, expressed in terms of α and β values, has not been clearly elucidated, but the α/β ratio has been suggested to range from 4 to 10 [5]. In one study, the α/β ratio of locally advanced pancreatic cancer was estimated to be 9.5, indicating relatively low fractionation sensitivity [6]. A recent systematic review of in vitro studies showed a mean surviving fraction at 2.0 Gy (SF2) of 0.48 ± 0.11 and 0.57 ± 0.13 in case of proton and photon radiation, respectively [7]. Current research is heading towards predicting the biological effect of radiation by referring to the tumor’s genetic characteristics [8].

The management of pancreatic cancer largely depends on its resectability. In case of resectable or borderline resectable tumors, offering neoadjuvant chemoradiotherapy (CRT) is suggested. The long-term results of the Dutch randomized PREOPANC trial, published recently in 2022, showed a significant improvement in overall survival (OS) with neoadjuvant CRT compared to upfront surgery (5-year OS rate: 20.5% vs. 6.5%, respectively) [9]. After surgical resection, the standard of care is to offer adjuvant chemotherapy [10], with conflicting evidence on the role of adjuvant radiation therapy [11,12]. As for unresectable tumors, data are still limited, but radiation may play a role in delaying progression of locally advanced disease or in relieving symptoms of pain, bleeding, or local obstruction [4]. Our group has shown that ablative radiation therapy following multiagent induction therapy for locally advanced pancreatic cancer was associated with durable locoregional control and favorable survival. Respective 12- and 24-month overall survival were 74% and 38%. Twelve- and 24-month cumulative incidence of locoregional failure were 17.6% and 32.8% respectively [13]. 

The radiation modalities currently recommended in the National Comprehensive Cancer Network (NCCN) Guidelines are photon-based, and protons are not considered standard of care [4]. Over the past two decades, considerable radiotherapy advances have been brought forward. Since the early 2000s, intensity-modulated radiotherapy (IMRT) has been applied in pancreatic cancer management, leading to reduced gastrointestinal toxicity compared to 3-dimensional conformal radiotherapy (3D-CRT) [14]. This has allowed for the intensification of prescription doses, thus reaching levels with ablative potential [14]. The application of proton therapy in the management of gastrointestinal cancers has been proposed to further decrease toxicity, given that protons, as mentioned above, have no exit dose and minimal scatter to the nearby organs at risk [3,15]. Evidence for using protons for pancreatic cancer is growing, as even more centers adopt this advanced radiation modality [14,16]. We present in this paper an up-to-date review of the literature on proton radiation treatment for pancreatic cancer.

## 2. Methods

This paper is a narrative review. We relied on PubMed to find past literature, with the search terms “proton therapy” and “pancreatic cancer”. The search results were scanned by title and abstract, and relevant articles were included. To ensure comprehensiveness, we also scanned the references and the list of “cited by” papers for each of the selected articles. No specific exclusion criteria were applied, and all original research articles were considered, including peer-reviewed papers and meeting/poster viewing abstracts. As for the prospective studies, we relied on www.clinicaltrials.gov (accessed on 29 April 2022) with “pancreatic cancer” under “condition or disease” and “proton therapy” under “other terms”. The search results were again scanned to select for relevant, ongoing trials. The findings of our search were then narrated in the review paper as per the type of study (dosimetric, clinical, or prospective trial) and the topics covered.

## 3. Dosimetric Data

One of the earliest studies to evaluate the dosimetric benefit of proton therapy in pancreatic cancer was published by Zurlo et al. in the year 2000 [17]. With a dose of 50 Gy and a boost of 20 Gy, spot scanning proton plans in two inoperable pancreatic cancer cases had improved conformity and dose homogeneity with no increase in morbidity when compared to 3D-CRT and IMRT. Another early dosimetric study, comparing protons to 3D-CRT, also showed reduced doses to the organs at risk (OARs), most significantly the left kidney and the spinal cord [18]. 

The only other study to have used a dose as high as the one used by Zurlo and colleagues was carried out by Bouchard et al. [19]. With a dose escalated to 72 Gy and a planning target volume (PTV) that was virtually translated into 11 different locations within the pancreas, protons were shown to be superior to photons in terms of normal tissue sparing, especially when the tumor was located just behind the small bowel.

Subsequent studies would evaluate the use of proton radiotherapy, while specifically looking into different settings. The application of proton therapy in the neoadjuvant setting was first evaluated by Kozak et al. from Massachusetts General Hospital [20]. Forming the basis for a subsequent clinical trial, this dosimetric study showed that a hypofractionated schedule (5 CGE × 5 fractions) of proton therapy is feasible. Neoadjuvant proton irradiation was also shown to be feasible by Lee et al., who also covered high risk nodal targets [21]. All patients in the previous study had tumors in the pancreatic head and were irradiated to a standard dose of 50.4 CGE.

Moving on to the adjuvant setting, Nichols et al. retrospectively generated proton plans for patients already treated with IMRT post-resection at the University of Florida [22]. While both plans adequately covered the target with 50.4 Gy, proton plans spared the OARs to a better degree, with a significantly reduced V20 to the small bowel (15.4% vs. 47.0%, *p* = 0.0156) and stomach (2.3% vs. 20.0%, *p* = 0.0313). Taking it one step further, Ding et al. specified that proton therapy planned with the modulated scanning technique reduces the doses to the OARs even more so than plans with the passive scattering technique, but both proton techniques achieved plans that were superior to the photon plans (IMRT, volumetric-modulated arc therapy (VMAT), and 3D-CRT) [23]. 

The same comparisons have been made for unresectable tumors. For masses in the pancreatic head, when compared to IMRT, proton therapy decreased radiation exposure to the kidneys, liver, and spinal cord [24] as well as the stomach, duodenum, and small bowel [25]. Of note, the decreased exposure to the last three OARs mentioned only applied to the low-dose regions (e.g., V20Gy, *p* < 0.01), and doses in the mid-high regions were actually increased. For example, the V45Gy of the stomach was 3.9% with IMRT versus 5.8% and 4.5% for proton double scattering and pencil beam scanning techniques, respectively (*p* < 0.05). The stomach and duodenum also proved to be dosimetrically problematic for Stefanowicz et al. who applied intensity-modulated proton therapy (IMPT) with simultaneous integrated boost (SIB) using either a 2- or 3-beam approach [26]. Neither one, however, met the dose constraints for the aforementioned OARs. In a later study, the same group compared IMPT to IMRT (VMAT/Tomotherapy (TOMO)) with SIB and noted superior sparing with IMPT for doses below 50 Gy [27]. In another comparison, when robust IMPT was applied instead, the stomach and duodenum were further spared over both low and high dose ranges [28].

As for the different planning techniques used within proton therapy, they were dosimetrically compared as well. Thompson et al. noted that pencil beam scanning (PBS) showed dosimetric superiority or at least equivalence in comparison to double scattering (DS) [25]. Later, Chuong and colleagues reported similar findings [29]. PBS showed improved target conformality and decreased the dose to several OARs, including the bowels and the spinal cord (e.g., median dose to 0.1 cc of the bowel space: 61 vs. 62.6 Gy (RBE), *p* = 0.001). Though statistically significant, the authors questioned how clinically relevant such a decrease was.

Multiple factors are to be considered in virtue of the location of the pancreas, one of which is the change in large bowel contents. It was shown that when the bowel contents were made more gaseous, there was a significant change in clinical target volume (CTV) coverage and the doses received by the spinal cord [30]. Another important consideration is the inter-fractional motion of the tumor. This was manifested as a significant reduction in target coverage in a comparison between proton and photon plans [31]. To mitigate such effects, a team from the German Cancer Research Center suggested using 4D magnetic resonance imaging (MRI) for planning as well as gating and 28-day fractionation for treatment [32,33]. They came to this conclusion by carrying out a longitudinal analysis of the interplay effect (between motion and coverage) and noted that such an effect decreases considerably at 28 fractions. Therefore, in case of hypofractionated treatments, such as stereotactic body proton therapy (SBPT), it becomes more imperative to reduce motion changes. Of note, it is not common to usually use MRI for proton planning. The authors suggested it as an imaging modality in virtue of the high soft tissue contrast that it offers at no extra imaging dose to the patients, thus allowing repeated images to be taken. Using time-resolved volumetric MRI (4D MRI) would provide precise information about motion along the different fractions.

The feasibility of SBPT was evaluated by Sio et al. after either single field or multiple field non-robust optimization (SFO/MFO) [34,35]. The authors noted that both target coverage and normal tissue sparing were affected by the range uncertainties and motion [34], and that the optimal compromise between them occurred with the SFO plan, using a 6 mm spot size and at a 5 mm optimization target volume (OTV) expansion [35]. The location of the tumor seems to be significant too. Comparing SBRT-SIB and VMAT, Liu et al. showed that 2-field proton SBPT was safer for tumors in the head, while photon VMAT was safer for tumors in the pancreatic body [36].

The limited range of protons is a double-edged sword [37]. On the one hand, it offers a sharp dose fall-off and zero exit dose for organ protection. On the other hand, it suffers from range uncertainty and biological enhancement effect at the distal end of the beam, resulting in unexpected dosimetric deviations to target and distal organs. Uncertainty originates from CT calibration and remains the major contributor to the range uncertainty for proton therapy [38]. When it goes unmanaged properly, range uncertainty may result in significant target coverage loss (under-range) or overdose to the OARs (over-range) at the distal end of the field. Range uncertainty margins, considering the heterogeneity of the beam path and often calculated as 3–5% of the water-equivalent beam path length, are routinely used in the clinic during proton planning [39]. Efforts have been put into reducing the planning margins and range uncertainties and further optimizing the dose to organs at the distal end of the target [40]. Studies have shown the application of dual-energy CT (DECT) to achieve a range uncertainty of 2% (versus 3.5%) for brain and prostate cancer patients [41]. However, most DECT studies are based on static imaging or phantom. Beam path variations induced by respiratory motion and daily anatomy changes are often observed in 4DCT evaluation for pancreatic patients [42]. Special caution should be given to consideration for reducing the range uncertainty margin in proton planning, especially for a case involving a moving target and/or an unstable beam path. Studies have shown a practical and effective way to reduce the bowel doses for a pancreatic cancer patient by injecting a spacing gel to increase the separation between tumor and bowel [43]. Realizing the lag of advanced range measurement tools in proton therapy, researchers have been working on range probing and verification [44]. However, no commercial systems are available yet in the clinic. Plan optimization considering 4D dynamic CT and relative biological effectiveness (RBE) enhancement has recently attracted the attention of researchers and can benefit pancreatic cancer patients for proton therapy. 4D plan optimization based on multiple CT phases improves the robustness of the treatment to respiratory motion [45]. Linear energy transfer (LET)-guided optimization in IMPT optimizes the dose and LET simultaneously to improve the treatment safety and robustness [46]. 

## 4. Clinical Data

The studies presented above were dosimetric data and their clinical implications are yet to be proven. 

The pilot clinical study in applying proton therapy for locally advanced pancreatic cancer was carried out by Kamigaki et al. in 2008 at the Hyogo Ion Beam Medical Center, Japan [47]. Eighteen patients underwent radiation alone at a dose of 40–50 GyE. Treatment was well tolerated, with only one patient sustaining a Grade 3 (G3) acute duodenitis. Out of the 18 patients, 12 had stable disease at follow-up while the others progressed. Four patients underwent surgical exploration, with one having an R0 resection. Figure 1 summarizes the timeline of clinical studies on proton therapy in the management of pancreatic cancer. 

A few years later, Hong and colleagues reported the results of a phase I trial, evaluating the safety of hypofractionated, neoadjuvant CRT with capecitabine for localized, resectable pancreatic cancer [48]. With escalated dose levels and different schedules, they determined that offering 25 GyE in 5 daily fractions was feasible. They further demonstrated the safety of this regimen in a phase II trial, published 3 years later, highlighting a rate of Grade 3 toxicity at just 4.1% [49]. In evaluating toxicity further, Tseng et al. analyzed 47 patients overlapping with the aforementioned studies and found the radiation dose received by the stomach to be directly related to the development of nausea and vomiting (V3-V15GyE, *p* < 0.05) [50].

Hypofractionated protons CRT was also applied in the definitive setting for locally advanced tumors. Terashima et al. reported the results of a phase I/II prospective study, where 50 patients with pancreatic adenocarcinoma received variable doses of proton therapy, most commonly field-within-field-administered 67.5 GyE in 25 fractions, along with concurrent high dose gemcitabine [51]. The 1-year freedom from local progression and overall survival rates were 81.7% and 76.8%, respectively. Endoscopy was ordered for symptomatic patients only, and grade 3+ gastric ulcer or hemorrhage were noted in 10% of the patients. The same group of researchers later reported the outcomes of 91 patients who were treated with the same regimen and who underwent endoscopy regardless of symptoms. This time, treatment-induced ulcers in the stomach and duodenum were noted in almost 50% of the patients, but without hemorrhage or perforation [52]. Such high rates of toxicity raised some questions among the radiation oncology community. Nichols et al., however, noted that these rates were most likely linked to the aggressive nature of the regimen, whether the high dose of radiation per fraction or the high dose of chemotherapy, rather than the administration of protons, per se [53]. Of note, a similar intervention was carried out in another study, but the resultant toxicity profile was significantly milder. Eighteen patients with stage 3 pancreatic cancer received a hypofractionated, field-in-field proton treatment with a dose as high as 60 GyE in 20 fractions to the gross tumor volume (GTV) concurrently with S-1 chemotherapy (Tegafur (pro-drug of 5-Fluorouracil), Gimeracil, and Oteracil potassium). Only 1 patient developed an acute, grade 3 gastric ulcer, and no late grade 3 gastrointestinal toxicities were noted. The patients also had promising outcomes, with 1-year local control and overall survival rates of 100% and 80%, respectively [54].

Researchers from the Hyogo Ion Beam Medical Center in Japan further expanded on the toxicity and survival outcomes of concurrent protons and Gemcitabine as first suggested above by Terashima and colleagues. In an attempt to decrease toxicity and improve doses received by the tumor, Lee et al. evaluated pancreatic cancer patients after spacer placement between the tumor and the gastrointestinal tract [55]. Dosimetric comparisons revealed that with a prescription dose of 67.5 GyE in 25 fractions, the tumor received higher doses postoperatively while also respecting normal tissue constraints. Out of the nine patients, one had a grade 4 gastric perforation, and two had grade 2 gastrointestinal ulcers. The resultant 1-year overall survival and local control rates were 50% and 100%, respectively. More recently, Ogura et al. investigated the factors associated with long-term survival for patients receiving this regimen [56]. In their cohort of 123 patients with nonmetastatic locally advanced pancreatic cancer, the 2-year OS was 35.7%. On multivariate analysis, they found longer survival rates to be associated with tumor localization in the pancreatic body/tail (*p* = 0.04) and absence of anterior peri-pancreatic invasion (*p* = 0.015).

Other fractionation schedules were also assessed with protons. Conventional fractionation was first reported by Nichols et al. in a study where 22 patients received 50.4–59.4 Gy concurrently with capecitabine and tolerated it well [57]. Grade 2 gastrointestinal toxicities were noted in three patients, with no other higher grade gastrointestinal toxicities. While the latter study assessed patients with pancreatic cancer at different stages (either resected, marginally resectable, or unresectable), another phase II trial focused on those with unresectable disease [58]. After receiving 59.4 Gy at 1.8 Gy per fraction, the patients demonstrated a 2-year freedom from local progression and overall survival rate of 69% and 31%, respectively. None sustained grade 2 or higher gastrointestinal toxicities. Furthermore, Jeshwa et al. also reported favorable toxicity profiles in 13 patients undergoing IMPT (45–50 Gy in 25 fractions) with concurrent capecitabine or 5-FU [59]. None of the patients sustained grade 3 or higher treatment-related adverse events, and there was no change in patient-related outcome questionnaire scores before and after treatment.

Some patients with initially unresectable disease receiving concurrent chemoradiotherapy eventually became eligible for surgery. In one retrospective chart review, six patients underwent surgery after receiving definitive proton therapy at a dose of 59.4 Gy concomitantly with capecitabine [60]. Out of the six patients, five were eligible for resection, two of which were R0 with minimal residual disease on pathology. Within 30 days of discharge, three patients were readmitted for the management of wound infection, delayed gastric emptying, or ischemic gastritis. In another single arm phase II trial, 49 patients with locally advanced, unresectable pancreatic cancer received eight cycles of FOLFIRINOX and losartan, after which they were restaged [61]. If the tumor was deemed radiographically resectable, patients received short course CRT (5 GyE × 5 + capecitabine). In case of persistent vascular involvement, patients received a longer course of CRT instead (50.4 Gy + vascular boost to 58.5 Gy + capecitabine or 5-FU). As a result, 34 out of 49 patients underwent resection, with an R0 resection rate of 61%. 

Proton therapy was also applied in the adjuvant setting. On review of a multicenter registry (Proton Collaborative Group), Nichols et al. reported the outcomes on 18 patients with resected pancreatic cancer status post adjuvant proton therapy at a median dose of 50.5 Gy [62]. At the time of surgery, the margins were either negative (6), close (8), or positive (4). The regimen was relatively well tolerated, and the 2-year survival rate was 37%. 

A proportion of patients recur locally after surgery. A retrospective chart review noted that proton therapy for these cases may be a viable option that offers good local control [63]. Out of 30 patients who locally recurred and subsequently underwent proton therapy, 23 progressed, but only 9 of them did so locally. The median local progression-free survival (LPFS) was 41.2 months, significantly longer than the median progression-free survival (PFS) of 12.3 months. Boimel and colleagues took it one step further and looked into the feasibility of reirradiation in recurrent pancreatic cancers [64]. In their study, 15 patients, who had received proton therapy at least 3 months prior, underwent proton therapy yet again to their locally recurrent pancreatic tumors. Reirradiation was well tolerated, with a rate of acute grade 3+ toxicity of 13%. The 1-year OS and LPFS were 67% and 72%, respectively. Of note, patients who concurrently received chemotherapy with proton reirradiation showed a higher median survival than those who did not.

The dosing of proton therapy and the feasibility of simultaneous integrated boosts (SIB) were assessed in several studies. Arimura et al. retrospectively reviewed the charts of 82 patients with stage 3 pancreatic cancer who underwent either 67.5 Gy using the field-in-field technique or 50 GyE along with chemotherapy [65]. The median OS and PFS were 22 and 15 months, respectively, with no difference between the two doses. Another study suggested otherwise. Comparing 54.0–67.5 GyE to 50 GyE concurrently with gemcitabine or S-1 chemotherapy, Hiroshima et al. noted that higher radiation doses were associated with higher overall survival (*p* = 0.015) and local control (*p* = 0.023) [66]. No severe gastrointestinal toxicities were reported. Kim et al. also showed the safety of SIB with protons [67]. The authors first defined two PTVs: PTV1 (Internal target volume (ITV) + 3–5 mm) and PTV2 (ITV + 7–12 mm), with a prescribed dose of 45 GyE to the first and 30 GyE to the second to be delivered along with chemotherapy. The mean OS and LPFS were 75.7% and 64.8%, respectively, and there was no grade 3+ toxicities. It was noted that those who had received induction chemotherapy had a significantly higher median overall survival (21.6 vs. 16.7 months, *p* = 0.031). The same group of researchers later reviewed patient outcomes with the same aforementioned radiation treatment (RT) but with different chemotherapy schedules: RT without upfront or maintenance (group 1), RT followed by maintenance (group 2), and RT with upfront chemotherapy (group 3) [68]. The median OS times were 15.3, 18.3, and 26.1 months, respectively (*p* = 0.043). Of note, there was a significantly higher number of older patients in group 1, followed by groups 2 and 3, and age was found to be an independent prognostic factor associated with OS. The treatment was safe and tolerable, with no grade 3+ acute or late toxicities due to radiation therapy.

While several studies investigated the role proton therapy in the treatment of pancreatic cancer as reviewed above, few compared it with the photon modalities. The first comparison was carried out by Lukens et al. at the University of Pennsylvania [69]. Thirteen patients with pancreatic adenocarcinoma, the majority of whom were staged as T3, received either proton therapy or photon therapy (3D-CRT/VMAT) at a dose of 54 Gy along with concurrent chemotherapy (5-FU/capecitabine). While the proton plans resulted in much less dose received by the stomach (median gastric V20: 45 vs. 102 cc3, *p* = 0.02), they did not result in a statistically significant decrease in the rate of grade 3 acute gastrointestinal toxicity (8% vs. 24%, *p* = 0.36). Another comparison by investigators from the same institution showed similar results even though more patients were included [70]. In a retrospective cohort of 105 patients who received adjuvant chemoradiation, the rate of acute grade 3 gastrointestinal toxicity was 5% in those who received protons versus 18% in those who received photons (*p* = 0.079). The mean number of hospitalizations was also insignificantly decreased in the proton group (0.95 vs. 1.31, *p* = 0.276), and there was no difference in overall survival. It is unclear how big the true decrease should at least be or, alternatively, how big the population should be in order to detect significance. Maemura and colleagues later reported similar findings [71]. In comparing proton therapy to a hyperfractionated accelerated photon therapy (HART) in locally advanced and unresectable disease, the authors found no survival advantage to using protons. In fact, HART led to a significantly higher tumor reduction rate (29.9% vs. 1.6%, *p* < 0.05). Otherwise, there was no significant difference in the median time to progression or median OS time. That being said, the toxicity profile during radiation treatment was different between the two modalities. While none of the patients undergoing HART therapy had gastrointestinal ulcers, 2 out of 10 of those undergoing proton therapy had grade 2 or grade 3 ulcers. Conversely, none of the proton patients sustained grade 3 or higher hematological toxicities, whereas 4 out of 15 of the HART patients did, with 3 of them sustaining grade 3 leukopenia. Hematological toxicity is a significant consideration with regards to oncologic outcomes. In what pertains to pancreatic cancer, it has been shown that increased irradiation of the vertebral bodies and the spleen is associated with the development of grade 2 and higher lymphopenia [72]. When severe enough (total lymphocyte count < 500 cells/mm^3^), lymphopenia is, in turn, associated with worse survival (HR: 2.879, *p* = 0.001) [73]. This has also been shown in other gastrointestinal cancers, namely esophageal [74] and hepatocellular cancer [75]. Table 1 summarizes the studies that compared photons and protons radiation therapy for the treatment of pancreatic cancer. 

## 5. Prospective Trials

For the time being, there are no randomized trials comparing proton and photon modalities for pancreatic cancer, and no such trials exist in the pipeline so far. What exists, instead, are several phase I/II single arm trials. The Proton Collaborative Group is currently assessing the overall survival of patients with borderline resectable, resectable, or unresectable tumors undergoing concomitant chemo–proton therapy, at an escalated dose with elective nodal irradiation [76]. Another group at Georgetown University Medical Center is looking into different PBT schedules concurrent with mFOLFIRINOX as an adjuvant treatment post-resection in a non-randomized, phase I trial called Proton-PANC [77]. Other trials are evaluating the PFS after neoadjuvant photon or proton RT concurrent with capecitabine and hydroxychloroquine [78], the maximum tolerated dose of gemcitabine and nab-paclitaxel concurrent with hypofractionated IMPT for locally advanced tumors [79], and the toxicity and possible perioperative complications after concurrent chemo–proton therapy [80]. 

Figure 2 summarizes the above studies on proton therapy in the management of pancreatic cancer, stratified by their special focus and time of publication. 

## 6. Discussion

Since Rutenberg et al.’s review article on proton beam therapy for pancreatic cancer [14], multiple studies have been published, upon which we expanded herein. Recent dosimetric studies compared more advanced radiation plans, such as IMPT vs. TOMO [27], robust IMPT vs. IMRT [28], and proton-based SBRT-SIB vs. VMAT [36]. Researchers have also looked further into the effects of changes in large bowel content [30] and ways at accounting for interfractional motion of the tumor [32,33]. Recent clinical studies explored the factors associated with long term survival [56] and the possible benefits of spacer placement [55], dose escalation [66], or different chemoradiotherapy schedules [68]. Others assessed the possible role of protons in locally recurrent tumors [63] or in initially unresectable tumors; most of which then became resectable [61]. In addition, two new trials have been started, with one looking into different proton therapy schedules given concurrently with mFOLFIRINOX [77] and the other assessing the feasibility of a preoperative chemo–proton therapy regimen for borderline resectable tumors [80].

Other significant results that have been published since Rutenberg et al.’s review are those of the Dutch randomized PREOPANC trial [9]. Those long-term results underscore survival benefit of administering chemoradiation therapy prior to surgical resection of pancreatic cancer. This encourages further research to be carried out to understand the role of radiation more clearly, whether photon or proton, in treating pancreatic tumors at different stages.

At our center, we have not been using proton RT consistently to treat patients with pancreatic cancer. We have been treating patients with unresectable or borderline resectable pancreatic tumors with ablative doses of photon radiation treatment to a biologic effective dose BED > 100 Gy. We treat those patients using deep-inspiration breath hold, and image-guided therapy matching to fiducial markers. We use a simultaneous integrated boost to treat to two dose levels that would encompass both the gross disease, and the microscopic disease [13,81,82]. Some of the regimens we have been using are 5 fractions to a total dose of 25/50 Gy using the Magnetic Resonance Imaging Guided Linear Accelerator (MRI-LINAC), and 15 fractions to 37.5/67.5 Gy and 25 fractions to 45/75 Gy on the linear accelerator. We have shown that ablative doses of photon RT was associated with durable locoregional control and favorable survival. Combining those techniques with proton radiation treatment is still work in progress. 

At times, there can be less dose conformity with protons vs. IMRT due to range uncertainty and fewer beams being used in proton planning. Typically, posterior or posterior oblique fields are used for proton therapy for pancreatic tumors, which place the luminal gastrointestinal organs at the distal end of the fields. Protons are associated with range uncertainties and enhanced biological effects at the end of the beam, which has the potential to lead to unexpected higher bowel doses and a higher risk of complications (bleeding, fibrosis, perforation, etc.). 

## 7. Conclusions

Applications of proton therapy in pancreatic cancer, whether dosimetric or clinical, have included different settings, ranging from neoadjuvant chemoradiation for resectable disease to definitive management of unresectable cancers. Radiation treatment planning can be challenging, because of the tumor motion uncertainties and the close proximity of pancreatic tumors to surrounding organs at risk (mainly bowel). This is especially true with proton RT considering the range uncertainty and fewer beams being used in proton planning compared to IMRT. Non-randomized comparative data have not consistently shown a statistical difference between proton and photon irradiation in terms of oncologic outcomes or toxicity to date, and randomized data so far are lacking. Further studies are also required to optimize fractionation schedules and concurrent chemotherapeutic options, while balancing efficacy and safety concerns. At our center, we have been treating patients with unresectable or borderline resectable pancreatic tumors with ablative doses of IMRT and using the MRI-LINAC for selected cases. 

## Figures and Tables

**Figure 1 cancers-14-02789-f001:**
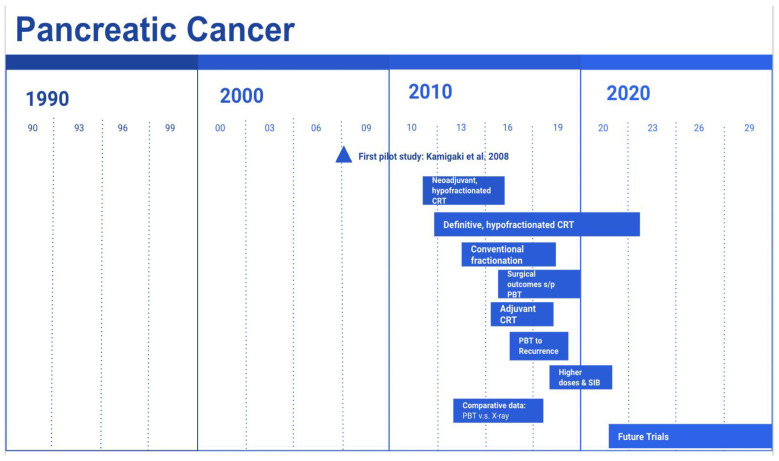
Timeline of clinical studies on the use of proton therapy in the management of pancreatic cancer.

**Figure 2 cancers-14-02789-f002:**
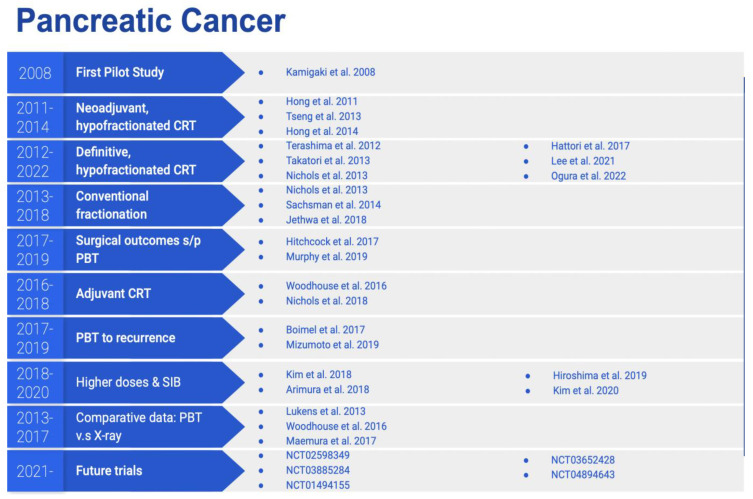
Studies on proton therapy for the treatment of pancreatic cancer, stratified by their special focus and time of publication.

**Table 1 cancers-14-02789-t001:** Studies summarizing outcomes and toxicities associated with the use of proton versus photon radiation therapy for the treatment of pancreatic cancer.

Authors	Year	Study Design	N	Comparison	OS	PFS	Toxicity
Lukens et al.	2013	Prospective	13				Grade 3
				PBT	NA	NA	8%
				3D-CRT or VMAT	NA	NA	24%
				*p*-value			0.36
Woodhouse et al.	2016	Retrospective	105				Grade 3 ^1^
				Proton	NA	NA	5%
				Photon	NA	NA	18%
				*p*-value			0.079
Maemura et al.	2017	Prospective	25		Median OS	TTP ^2^	Grade 3 ^3^
				Proton	22.3 months	15.4 months	No hematological 1/10 non-heme
				Photon (HART)	23.4 months		4/15 hematological No non-heme
				*p*-value	N.S.	N.S.	NA

^1^ Acute grade 3 gastrointestinal toxicity, ^2^ TTP: median time to progression, ^3^ hematological versus non-hematological toxicity.

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
