# Peer review of "Proton Therapy in the Management of Pancreatic Cancer"

_cancers, 2022, doi:10.3390/cancers14112789_

Round 1
Reviewer 1 Report
Thanks to the authors for this thorough review. The article is well written and shows a proper literature research.
Find below some comments and suggestions:
It is not specified how the literature search was performed for this review paper. Was it done systematically, e.g. specifying some search terms in PubMed, or rather randomly? A clarification could be added on this to the paper to know the type. See:
https://libguides.csu.edu.au/review/Types
https://bmcmedresmethodol.biomedcentral.com/articles/10.1186/s12874-018-0545-3
https://libraryguides.griffith.edu.au/c.php?g=451351&p=3333115
Page 1:
- this radiation advanced --> this advanced radiation
Page 5:
- "they did not result in a statistically significant decrease in the rate of grade 3 acute gastrointestinal toxicity (8% vs. 24%, p=0.36)" --> Is this because the number of patients included in the cohort was not enough to draw statistically significant conclusions (13)? Maybe it can be mentioned here explicitly, not only on the table. Or specifying the power of the test. Alternatively, how big should the true 'decrease' at least be, so that it could be detected with significance with just 13 patients?
Since the authors cite a Review Paper (Rutenberg et al 2020) on the same topic, it would be also interesting to highlight what has changed since then, what are the new results or literature, to better identify the added value of the current manuscript. Or at least comment on these in the reply letter to the reviewers.
I think it would be helpful to add some more detailed comment on range uncertainties in proton therapy, and how these may affect when treating the cancer. There have been efforts to reduce safety margins in proton therapy by better characterizing the patient tissue, e.g. using DECT (https://pubmed.ncbi.nlm.nih.gov/34774653/). Are there efforts going on in this direction at your centre (Conclusions section), is it something that the community is pushing for? Or is this not expected to yield significant benefits, as organ motion might be more important? Would in vivo range verification help instead?
These related papers might also give some insights:
https://www.ncbi.nlm.nih.gov/pmc/articles/PMC7052758/
https://www.ncbi.nlm.nih.gov/pmc/articles/PMC8493097/
https://www.advancesradonc.org/article/S2452-1094(21)00115-9/pdf
Finally, in the context of proton therapy, I think it would be interesting to the reader to get more detailed description on the effect of lymphopenia, not just reporting the general toxicity. Is there some literature specific for pancreas and protons? Or can results from other GI cancer papers with proton beams be extrapolated to pancreas tumors? Maybe these papers can shed some light on it:
https://www.ncbi.nlm.nih.gov/pmc/articles/PMC6075713/
https://pubmed.ncbi.nlm.nih.gov/23648440/
https://www.ncbi.nlm.nih.gov/pmc/articles/PMC7937383/
https://mdpi-res.com/d_attachment/cancers/cancers-14-01167/article_deploy/cancers-14-01167.pdf?version=1645699013
https://ro-journal.biomedcentral.com/articles/10.1186/s13014-021-01969-1
Author Response
Thank you for the feedback.
Please find below a point-by-point response to the provided comments.
Point 1: It is not specified how the literature search was performed for this review paper. Was it done systematically, e.g. specifying some search terms in PubMed, or rather randomly? A clarification could be added on this to the paper to know the type. See:
https://libguides.csu.edu.au/review/Types
https://bmcmedresmethodol.biomedcentral.com/articles/10.1186/s12874-018-0545-3
https://libraryguides.griffith.edu.au/c.php?g=451351&p=3333115
Response 1: This paper is a narrative review. We relied on PubMed to find past literature, with the search terms “proton therapy” and “pancreatic cancer.” The search results were scanned by title and abstract, and relevant articles were included. To ensure comprehensiveness, we also scanned the references and the list of “cited by” papers for each of the selected articles. No specific exclusion criteria were applied, and all original research articles were considered, including peer-reviewed papers and meeting/poster viewing abstracts. As for the prospective studies, we relied on www.clinicaltrials.gov with “pancreatic cancer” under condition or disease and “proton therapy” under other terms. The search results were again scanned to select for relevant, ongoing trials. The findings of our search were then narrated in the review paper as per the type of study (dosimetric, clinical, or prospective trial) and the topics covered. This was added in a separate section of the paper, called “Methods.”
Point 2: Page 1:
- this radiation advanced --> this advanced radiation
Response 2: Suggested edit was done.
Point 3: Page 5:
- "they did not result in a statistically significant decrease in the rate of grade 3 acute gastrointestinal toxicity (8% vs. 24%, p=0.36)" --> Is this because the number of patients included in the cohort was not enough to draw statistically significant conclusions (13)? Maybe it can be mentioned here explicitly, not only on the table. Or specifying the power of the test. Alternatively, how big should the true 'decrease' at least be, so that it could be detected with significance with just 13 patients?
Response 3: The small number of patients (n=13) included in the study by Lukens et al. may have contributed to the results being statistically insignificant. However, in the study carried out later by the same group of researchers and which included an even larger cohort (n=105), the differences in toxicity were yet again insignificant. It is unclear how big the true “decrease” should at least be or, alternatively, how big the population should be in order to detect significance. The two aforementioned studies were published as poster viewing abstracts in the International Journal of Radiation Oncology and no in-depth discussion was included. In the revised draft, we made sure to explicitly mention the difference in cohort size and to highlight the uncertainty in detecting significance (see page 6).
Point 4: Since the authors cite a Review Paper (Rutenberg et al 2020) on the same topic, it would be also interesting to highlight what has changed since then, what are the new results or literature, to better identify the added value of the current manuscript. Or at least comment on these in the reply letter to the reviewers.
Response 4: We added a paragraph to section 5 [prospective trials], whereby we highlighted the new literature that has been published since Rutenberg et al.’s review, to better show the added value of our manuscript. New dosimetric studies mainly compared more advanced techniques or looked further into motion effects. New clinical studies further applied proton therapy in locally advanced tumors, while escalating the dose, placing a spacer between the tumor and other GI organs, comparing different chemotherapy schedules, or assessing the R0 resection rate in those tumors that eventually became resectable. Other new clinical studies assessed the factors associated with long term survival and applied proton therapy in locally recurrent tumors. We also highlighted the two trials that have since been started, which either look into different proton therapy schedules or the feasibility of a preoperative chemo-proton therapy regimen. The recent studies as well as those currently under way demonstrate that researchers are yet to optimize the application of proton therapy in pancreatic cancer.
Point 5: I think it would be helpful to add some more detailed comment on range uncertainties in proton therapy, and how these may affect when treating the cancer. There have been efforts to reduce safety margins in proton therapy by better characterizing the patient tissue, e.g. using DECT (https://pubmed.ncbi.nlm.nih.gov/34774653/). Are there efforts going on in this direction at your centre (Conclusions section), is it something that the community is pushing for? Or is this not expected to yield significant benefits, as organ motion might be more important? Would in vivo range verification help instead?
These related papers might also give some insights:
https://www.ncbi.nlm.nih.gov/pmc/articles/PMC7052758/
https://www.ncbi.nlm.nih.gov/pmc/articles/PMC8493097/
https://www.advancesradonc.org/article/S2452-1094(21)00115-9/pdf
Response 5:
We will add the paragraph below under dosimetric data section 3
The limited range of protons is a double-edged sword [1]. On the one hand, it offers a sharp dose fall-off and zero exit dose for organ protection. On the other hand, it suffers from range uncertainty and biological enhancement effect at the distal end of the beam, resulting in unexpected dosimetric deviations to target and distal organs. Uncertainty originated from CT calibration and remains the major contributor to the range uncertainty for proton therapy [2]. All the treatment sites and proton delivery techniques were affected. When it goes unmanaged properly, Range uncertainty may result in significant target coverage loss (under-range) or overdose to the OARs (over-range) at the distal end of the field. Range uncertainty margins, considering the heterogeneity of the beam path and often calculated as 3-5% of the water-equivalent beam path length, are routinely used in the clinic during proton planning [3]. Efforts have been put into reducing the planning margins and range uncertainties and further optimizing the dose to organs at the distal end of the target [4]. Studies have shown that the application of dual-energy CT achieved a range uncertainty of 2% (versus 3.5%) for brain and prostate cancer patients [5]. However, most DECT studies are based on static imaging or phantom. Beam path variations induced by respiratory motion and daily anatomy changes are often observed in 4DCT evaluation for pancreatic patients [6]. Special caution should be given to consideration for reducing the range uncertainty margin in proton planning, especially for a case involving a moving target and/or an unstable beam path. Instead, studies have shown a practical and effective way to reduce the bowel doses for a pancreatic cancer patient by injecting a spacing gel to increase the separation between tumor and bowel [7]. Realizing the lag of advanced range measurement tools in proton therapy, researchers have been working on range probing and verification [8]. However, no commercial systems are available yet in the clinic. Plan optimization considering 4D dynamic CT and RBE enhancement recently attracted the attention of researchers and can benefit pancreatic cancer patients for proton therapy. 4D plan optimization based on multiple CT phases improves the robustness of the treatment to respiratory motion [9]. LET-guided optimization in intensity-modulated proton therapy optimizes the dose and LET simultaneously to improve the treatment safety and robustness [10].
Reference:
1. Paganetti H. Range uncertainties in proton therapy and the role of Monte Carlo simulations. Phys Med Biol. 2012 Jun 7;57(11):R99-117. doi: 10.1088/0031-9155/57/11/R99. Epub 2012 May 9. PMID: 22571913; PMCID: PMC3374500.
2. Li B, Lee HC, Duan X, Shen C, Zhou L, Jia X, Yang M. Comprehensive analysis of proton range uncertainties related to stopping-power-ratio estimation using dual-energy CT imaging. Phys Med Biol. 2017 Aug 9;62(17):7056-7074. doi: 10.1088/1361-6560/aa7dc9. PMID: 28678019; PMCID: PMC5736379.
3. Yang M, Zhu XR, Park PC, Titt U, Mohan R, Virshup G, Clayton JE, Dong L. Comprehensive analysis of proton range uncertainties related to patient stopping-power-ratio
estimation using the stoichiometric calibration. Phys Med Biol. 2012 Jul 7;57(13):4095-115. doi: 10.1088/0031-9155/57/13/4095. Epub 2012 Jun 7. PMID: 22678123; PMCID: PMC3396587.
4. Han D, Hooshangnejad H, Chen CC, Ding K. A Beam-Specific Optimization Target Volume for Stereotactic Proton Pencil Beam Scanning Therapy for Locally Advanced Pancreatic Cancer. Adv Radiat Oncol. 2021 Jul 29;6(6):100757. doi: 10.1016/j.adro.2021.100757. PMID: 34604607; PMCID: PMC8463829.
5. Peters N, Wohlfahrt P, Hofmann C, Möhler C, Menkel S, Tschiche M, Krause M, Troost EGC, Enghardt W, Richter C. Reduction of clinical safety margins in proton therapy enabled by the clinical implementation of dual-energy CT for direct stopping-power prediction. Radiother Oncol. 2022 Jan;166:71-78. doi: 10.1016/j.radonc.2021.11.002. Epub 2021 Nov 11. PMID: 34774653.
6. Tryggestad EJ, Liu W, Pepin MD, Hallemeier CL, Sio TT. Managing treatment-related uncertainties in proton beam radiotherapy for gastrointestinal cancers. J Gastrointest Oncol. 2020 Feb;11(1):212-224. doi: 10.21037/jgo.2019.11.07. PMID: 32175124; PMCID: PMC7052758.
7. Rao AD, Feng Z, Shin EJ, He J, Waters KM, Coquia S, DeJong R, Rosati LM, Su L, Li D, Jackson J, Clark S, Schultz J, Hutchings D, Kim SH, Hruban RH, DeWeese TL, Wong J, Narang A, Herman JM, Ding K. A Novel Absorbable Radiopaque Hydrogel Spacer to Separate the Head of the Pancreas and Duodenum in Radiation Therapy for Pancreatic Cancer. Int J Radiat Oncol Biol Phys. 2017 Dec 1;99(5):1111-1120. doi: 10.1016/j.ijrobp.2017.08.006. Epub 2017 Aug 14. PMID: 28943075; PMCID: PMC5699940.
8. Meijers A, Free J, Wagenaar D, Deffet S, Knopf AC, Langendijk JA, Both S. Validation of the proton range accuracy and optimization of CT calibration curves utilizing range probing. Phys Med Biol. 2020 Feb 4;65(3):03NT02. doi: 10.1088/1361-6560/ab66e1. PMID: 31896099.
9. Engwall E, Fredriksson A, Glimelius L. 4D robust optimization including uncertainties in time structures can reduce the interplay effect in proton pencil beam scanning radiation therapy. Med Phys. 2018 Jul 16. doi: 10.1002/mp.13094. Epub ahead of print. PMID: 30014478.
10. Liu C, Patel SH, Shan J, Schild SE, Vargas CE, Wong WW, Ding X, Bues M, Liu W. Robust Optimization for Intensity Modulated Proton Therapy to Redistribute High Linear Energy Transfer from Nearby Critical Organs to Tumors in Head and Neck Cancer. Int J Radiat Oncol Biol Phys. 2020 May 1;107(1):181-193. doi: 10.1016/j.ijrobp.2020.01.013. Epub 2020 Jan 25. PMID: 31987967.
Point 6: Finally, in the context of proton therapy, I think it would be interesting to the reader to get more detailed description on the effect of lymphopenia, not just reporting the general toxicity. Is there some literature specific for pancreas and protons? Or can results from other GI cancer papers with proton beams be extrapolated to pancreas tumors? Maybe these papers can shed some light on it:
https://www.ncbi.nlm.nih.gov/pmc/articles/PMC6075713/
https://pubmed.ncbi.nlm.nih.gov/23648440/
https://www.ncbi.nlm.nih.gov/pmc/articles/PMC7937383/
https://mdpi-res.com/d_attachment/cancers/cancers-14-01167/article_deploy/cancers-14-01167.pdf?version=1645699013
https://ro-journal.biomedcentral.com/articles/10.1186/s13014-021-01969-1
Response 6: We elaborated further on the toxicity profiles reported in the comparative study by Maemura et al. and which included hematological toxicities. We also made sure to highlight the association between lymphopenia and survival outcomes as previously shown for pancreatic tumors and other gastrointestinal cancers (see page 6).
Reviewer 2 Report
This review “Proton Therapy in the Management of Pancreatic Cancer” by Kobeissi et al summarizes the literature on the use of proton therapy in the management of
pancreatic cancer. The review is very well written and the compilation of the dosimetric and clinical data is very up-to-date. However, the review falls short in covering many aspects of the management of pancreatic cancer using radiation therapy. Following are some points to consider adding in the review:
- The author should consider writing a detailed statement or paragraph on the significance of the dosimetric and clinical data. Where this data/review could be helpful for future work on the management of pancreatic cancer?
- What are the benefits and practical shortcomings of radiation therapy in pancreatic cancer management?
- For the general audience, the author should consider adding a piece of brief background information on radiation therapy and/or pancreatic cancer.
- What are the different radiation therapies currently being used for pancreatic cancer? E.g. adjuvant and neoadjuvant radiation therapy, radiation therapy for different stages of pancreatic cancer and their outcome, etc.
- What is the advancement in the field of radiation therapy and data acquisition happened in this decade for pancreatic cancer treatment?
Author Response
Thank you for the feedback.
Please find below a point-by-point response to the provided comments.
Point 1: The author should consider writing a detailed statement or paragraph on the significance of the dosimetric and clinical data. Where this data/review could be helpful for future work on the management of pancreatic cancer?
Response 1: We added a paragraph to the section 5 to highlight the new literature that has been published since Rutenberg et al.’s review, to better show the added value of our manuscript.
Point 2: What are the benefits and practical shortcomings of radiation therapy in pancreatic cancer management?
Response 2: The survival benefits and the different uses of radiation therapy for pancreatic cancer are highlighted in the introduction (fourth paragraph). Toxicity is a major limitation as first mentioned in the introduction and elaborated further upon in the manuscript. We also made sure to highlight the uncertainties, whether in beam ranges or tumor motion, that are particularly significant considerations in proton irradiation of gastrointestinal tumors (section on dosimetric data). Equal access to protons is another practical shortcoming, in addition to motion management and image-guidance which are more readily available with photons RT as opposed to protons RT. Ablative radiation therapy [BED>100] to pancreatic tumors is safer with IMRT; protons are associated with a larger penumbra at the interface of the tumor with luminal GI organs which leads to higher bowel doses and higher risk of complications [bleeding, perforation, etc.]. At our institution, we have not been using protons RT to treat pancreatic tumors.
Point 3: For the general audience, the author should consider adding a piece of brief background information on radiation therapy and/or pancreatic cancer.
Response 3: The first paragraph of the introduction introduces pancreatic cancer and current survival outcomes. We added a short paragraph just after the first to elaborate more about photon and proton radiation for the general audience.
Point 4: What are the different radiation therapies currently being used for pancreatic cancer? E.g. adjuvant and neoadjuvant radiation therapy, radiation therapy for different stages of pancreatic cancer and their outcome, etc.
Response 4: The fourth paragraph of the introduction touches upon the different settings in which radiation therapy can be applied for pancreatic cancer. The neoadjuvant setting is particularly significant, given the recent long-term results of the PREOPANC trial, which showed significantly improved overall survival when comparing neoadjuvant chemoradiation to surgery alone. Evidence is not as clear, however, concerning adjuvant radiation. For unresectable tumors, radiation therapy delays progression and relieves symptoms. Our group has shown that ablative RT for locally advanced pancreatic cancer was associated with durable locoregional tumor control and favorable survival [PMID: 33704353].
Point 5: What is the advancement in the field of radiation therapy and data acquisition happened in this decade for pancreatic cancer treatment?
Response 5: A few lines on the advancement in radiation therapy for pancreatic cancer were added to the last paragraph of the introduction. We highlighted how IMRT has had an improved toxicity profile compared to 3D-CRT, which has allowed for intensification of doses to reach ablative levels. We then mentioned how proton therapy was suggested as a modality to further decrease toxicity. The review paper shall then narrate to the readers the advancements that have taken place in what pertains to proton therapy and pancreatic cancer. Ablative RT has certainly changed the paradigm of treatment for pancreatic cancer. Our group has shown that ablative RT for locally advanced pancreatic cancer was associated with durable locoregional tumor control and favorable survival [PMID: 33704353].
Reviewer 3 Report
This article is a classical review article summarizing the literature on the use of proton therapy in the multimodal treatment of patients with pancreatic cancer. It is very well written and could be used as a model for how such an article should be written. It is very educating, adequate, precise and contains useful remarks and a useful discussion. Since it reviews the use of a radiotherapeutic modality it deals with both "technical" and "clinical" data. When it comes to dosimetry the article discusses data with special respect to comparisons with what can be achieved with conventional photons, how to avoid undue spread to normal tissues in special clinical situations, for patients treated with with an adjuvant intention after radical surgery, for patients with unresectable tumours and patients with borderline resectable tumours in a neoadjuvant fashion. The points of dosimetry discussion with special emphasis on doses to duodenum and stomach are well carified with a discussion on how targets can differ when when localization of primary target in the pancreas is considered, spread to regional lymph nodes, and the prophylactic incorporation of othes tissues at risk are taken into consideration. The references in this section are well chosen and very relevant for the following discussion on clinical data. The authors conclude this section by summarizing that the clinical implications of the "technical" findings and the potential advantages which are presented remain to be proven.
The overview of clinical data is also very well presented with relevant, recent studies and completed with an educating figure 1, showing a timeline of clinical studies on the use of proton therapy for patients with pancreatic cancer. This section summarizes almost all relevant data concerning 1. given doses 2. different fractionation schedules 3. the addition of drugs during radiotherapy ("chemoradiotherapy") and 4. the impact of induction chemotherapy.
The authors present important conclusions of their own in this ambitious review under the titles "Prospective Trials" and "Conclusions".
MY OWN POINTS OF VIEW WHAT MIGHT BE ADDED AND ADDRESSED IN A FEW SENTENCES:
In either the section on "Dosimetry" or rather "Clinical Data" a few words on radiobiology could be addressed: What is known on the inherent radiosensitivity of pancreatic tumours, as measured in vitro, expressed as surviving fraction at 2.0 Gy (SF2) or otherwise? Do we know-or not-anything about the alfa/beta ratio of pancreatic tumour cells? Maybe could the authors very briefly comment the article "Pan-cancer prediction of radiotherapy benefit using genomic -adjusted radiation dose (GARD): a cohort-based pooled analysis", by Jacob G Scott, Geoffrey Sedor, Patrick Ellsworth et al. The importance of the results of data in reference 2 could be further stressed in the section of the concluding remarks.
But: The article is excellent.
Author Response
Thank you for the feedback.
Please find below a point-by-point response to the provided comments.
Point 1: In either the section on "Dosimetry" or rather "Clinical Data" a few words on radiobiology could be addressed: What is known on the inherent radiosensitivity of pancreatic tumours, as measured in vitro, expressed as surviving fraction at 2.0 Gy (SF2) or otherwise? Do we know-or not-anything about the alfa/beta ratio of pancreatic tumour cells?
Response 1: A short paragraph about radiobiology was added to the introduction (see third paragraph) to indicate that: the radiosensitivity of pancreatic tumor cells, expressed in terms of ⍺ and β values, has not been clearly elucidated, but the ⍺/β ratio was suggested to range from 4 to 10 (Jones 2020). In one study, the ⍺/β ratio for chemoradiation of locally advanced pancreatic cancer was estimated to be 9.5, indicating relatively low fractionation sensitivity (Prior 2018). A recent systematic review of in vitro studies showed a mean surviving fraction at 2.0 Gy (SF2) of 0.48 ± 0.11 and 0.57 ± 0.13 in case of proton and photon radiation, respectively (Wang 2022).
Point 2: Maybe could the authors very briefly comment the article "Pan-cancer prediction of radiotherapy benefit using genomic-adjusted radiation dose (GARD): a cohort-based pooled analysis", by Jacob G Scott, Geoffrey Sedor, Patrick Ellsworth et al.
Response 2: In the short paragraph we added about radiobiology, we briefly mentioned that genomics may be applied to predict the biological effect of radiation.
Point 3: The importance of the results of data in reference 2 could be further stressed in the section of the concluding remarks.
Response 3: We added a short paragraph to the section 5 to highlight the results of the PREOPANC trial.
Round 2
Reviewer 1 Report
Congratulations to the authors for their excellent work and perfect replies. They have significantly enhanced the manuscript's quality. It is ready for publication.
Just two minor comments:
419: "due to (...) fewer beams are used in proton planning." --> being used ?
422: "Protons are associated with respiratory motion" --> I do not understand the meaning of this part of the sentence.
Author Response
Thank you for your feedback. The proposed modifications have been done.
Reviewer 2 Report
The authors made appropriate changes in the manuscript as required.
Author Response
Thank you for your feedback!